# Automatic Construction of Evaluation Suites for Natural Language Generation Datasets

**Simon Mille**
Universitat Pompeu Fabra
simon.mille@upf.edu

**Kaustubh D. Dhole**
Amelia Science, IPsoft R&D
kdhole@ipsoft.com

**Saad Mahamood**
trivago N.V.
saad.mahamood@trivago.com

**Laura Perez-Beltrachini**
University of Edinburgh
lperez@ed.ac.uk

**Varun Gangal**
Carnegie Mellon University
vgangal@cs.cmu.edu

**Mihir Kale**
Google Research
mihirkale@google.com

**Emiel van Miltenburg**
Tilburg University
C.W.J.vanMiltenburg@uvt.nl

**Sebastian Gehrmann**
Google Research
gehrmann@google.com

## Abstract

Machine learning approaches applied to NLP are often evaluated by summarizing their performance in a single number, for example accuracy. Since most test sets are constructed as an i.i.d. sample from the overall data, this approach overly simplifies the complexity of language and encourages overfitting to the head of the data distribution. As such, rare language phenomena or text about underrepresented groups are not equally included in the evaluation. To encourage more in-depth model analyses, researchers have proposed the use of multiple test sets, also called challenge sets, that assess specific capabilities of a model. In this paper, we develop a framework based on this idea which is able to generate controlled perturbations and identify subsets in text-to-scalar, text-to-text, or data-to-text settings. By applying this framework to the GEM generation benchmark, we develop evaluation suites made of 80 challenge sets, demonstrate the kinds of analyses that it enables, and shed light onto the limits of current generation models.

## 1  Introduction

A very commonly used approach for assessing the performance and generalization of a given machine learning model is to compute its accuracy on a held-out dataset drawn i.i.d. from an underlying data distribution. This holds especially for datasets that are used to "benchmark" multiple models in an otherwise fixed environment. However, this regime has been criticized since a single performance number may hide numerous shortcomings of a model. These shortcomings appear, for example, when a model is presented with examples that occur less frequently in the data [1], noisy data that doesn't fully reflect the training distribution [2], or inputs that are robust to potential shortcuts a model may take [3]. In parallel, performance numbers are artificially inflated due to the usually high performance of the model on commonly seen examples.

It is thus necessary to more carefully construct test sets and go beyond the i.i.d. testing. However, it is challenging to re-release existing datasets with new splits that address these issues, since it breaks comparability with published numbers. An approach that combats the over-reliance on static test sets without introducing compatibility issues is to add additional test sets, thus creating an entire *evaluation suite*. An extension can, for example, manifest as small perturbations of test instances that

35th Conference on Neural Information Processing Systems (NeurIPS 2021), Sydney, Australia.

change the gold label [4], or connect perturbations to linguistic capabilities of model in NLP (e.g., negation or changing of entities) that lead to predictable changes in outputs [5]. These projects have inspired work on entire frameworks that help create evaluation suites, such as the RobustnessGym [6]. One commonality between these approaches is that they (mostly) study language classification tasks.

Unlike language classification, natural language generation (NLG) tasks, which we study in this work, have many correct outputs in an exponentially large search space. That means that when the model output changes in response to input perturbations, it is unclear whether the change was desired and whether it improved the generated text. Consequently, existing methods to construct and evaluate on *challenge sets* that assume deterministic changes to the reference or label of an example do not work for NLG. Our proposed solution to this problem is to closely tie in the challenge set construction process with an evaluation that aims to provide an in-depth characterization of model outputs.

To construct evaluation suites, we propose using a combination of linguistically informed perturbations, dataset-specific subsets of particular interest, and newly collected data. We release a framework called the *NL-Augmenter* that enables developers of text-to-text, data-to-text, and text-to-scalar datasets to produce evaluation suites and propose their own challenge set types. To demonstrate how we envision the output characterization to work for evaluation suites and as a case study of the expressiveness of our framework, we generated 80 challenge sets across 12 evaluation suites for the GEM benchmark [7]. We analyze the outputs of four different models and show that we can expose model behaviors that would remain hidden with "standard" test sets. Specifically, we show that (1) what it means for an input example to be difficult for a model depends strongly on the task, (2) i.i.d test sets overestimate the performance on realistic inputs that cover new topics, and (3) models are brittle to minor, but systematic, edits to inputs that would be easy to overcome as humans.

## 2  Background

**From leaderboards to evaluation suites** Benchmarks enable a comparison between systems in an otherwise static environment which can be a single dataset and an agreed-upon metric or a combination of challenges that measure specific model capabilities. In NLP, there has been a recent trend toward the latter, with benchmarks that include multiple datasets [8, 9, 10]. Even in this setting, the model is trained on each dataset individually and its performance typically summarized with a single metric computed over a single test set. This paradigm has been criticized since it encourages overfitting without regard to model shortcuts [11], and because it demotes considerations like fairness and model size [12]. Inspired by earlier research like that by Quiñonero-Candela et al. [13], we focus on the underlying data that is used to produce the summarizing numbers - specifically, the test sets.[1]

**Informative Data Splits** Relying on a single train-test split means that there is an inherent element of randomness that can influence the model performance. Gorman et al. [17] point out that it can lead to incorrect system rankings and thus advocate training on multiple random splits. However, Søgaard et al. [18] argue that random i.i.d. splits lead to overly optimistic performance estimates, and that taking a more informed approach like training on short sentences but testing on long ones, can be more informative when assessing a model's capability to generalize beyond seen examples. They thus advocate for multiple independent test sets and try multiple "informed splitting" approaches, e.g., based on publication date of a news article. There already exist multiple datasets with both i.i.d and constructed test sets, e.g., with unseen topics [19], unseen feature combinations [20], or splits based on linguistic properties of test examples [21]. Another approach is to delegate the slicing agency to those working with a dataset through interactive systems like Errudite [22] or the Language Interpretability Tool [23]. In this work, we aim for the middle ground between these two approaches by making it easier for dataset curators to generate multiple splits of their data.

**Additional Examples** While the identification of interesting subsets can enable powerful analyses, evaluation on subsets can only reveal information about examples that already exist in the test set in the first place. This excludes many real-world phenomena, such as time- or domain-shift [24], adversarial examples, and in many cases slices may be too small to yield meaningful results. There have been many different approaches to address this problem, the most prominent of which has been adversarial data collection. This can happen in a fully automated way by generating perturbations that models perform poorly on [25, 26, 4] or crowd sourcing data that looks ordinary but is challenging

---

[1]This work focuses on the investigation of task-specific models. Some works also investigate pretrained models with evaluation suites and aim to find out the linguistic features they can represent (e.g., [14, 15, 16]).

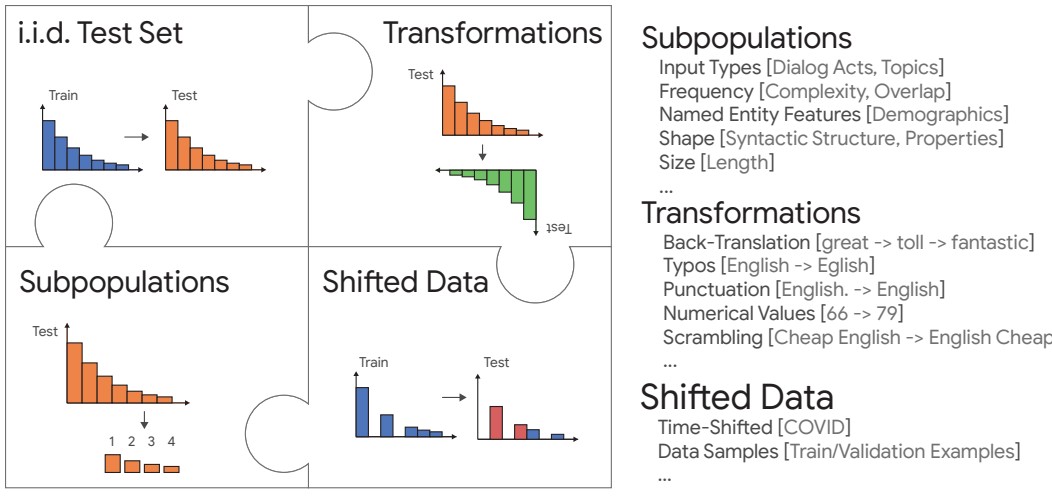

Figure 1: Illustration of the types of evaluation suites that can be constructed from a given dataset.

for models [27, 28, 29]. However, Bowman and Dahl [1] argue against adopting this fully adversarial regime. Adversarial approaches tend to conflate examples that are out-of-distribution and those that are underrepresented in the dataset. It is desirable to focus on methods that over-sample underrepresented examples instead of fully adversarial ones, which can be arbitrary and unrealistic.

There exists several similar approaches which focuses on natural language understanding tasks. For example, Ribeiro et al. [5] define linguistically motivated, plausible perturbations with deterministic output changes. In particular, a negation should flip a binary output of a sentiment classifier while changing out a named entity should not. McCoy et al. [3] developed a template-based dataset for natural language inference to test whether a model is using shallow heuristics. Tan et al. [30] connect the idea of over-sampling examples from a set of specific attributes to reliability testing. They argue that not only do we need to enumerate the worst case and the average performance per attribute we may want to evaluate. Taking inspiration from the idea to enumerate the performance of all attributes of interest, our framework connects attributes to appropriate data construction techniques. Conceptually, the work closest to ours is the concurrently introduced RobustnessGym [6] which uses the same three constructions (and additionally adversarial attacks), but mostly focus on the more narrow domain of natural language inference. Their investigation of summarization models is limited to the identification of subsets. Our framework for building datasets can be interpreted as a compatible extension of RobustnessGym to the generation domain, with more supported tasks.

## 3 Our Framework

### 3.1 Types of Challenge Sets

There have been many different approaches to the construction of targeted test sets, and almost as many terms for these approaches. We thus clarify the terminology and describe our methodology. We define a collection of test sets for a single corpus as its *evaluation suite*; an evaluation suite comprises multiple *challenge sets* which can be constructed in various ways. We distinguish between three challenge set categories, roughly following the categorization by Goel et al. [6]:

A **Subpopulation** is a subset of the existing test set in which examples share a commonality of interest (e.g., input complexity or topic). This category requires familiarity with a dataset and an understanding of its computational challenges. Our framework (see Section 3.2) assists in this process by keeping track of evaluation results for each subpopulation either by accessing cached overall results (for example-level metrics) or by recomputing them (for corpus-level metrics). A **Transformation** is a type of challenge set that modifies inputs and potentially the target text (e.g., by shuffling input order or modifying capitalization and punctuation). In our framework, we further define the **Transformation-Parent** set. Transformations bring about the need to compute additional system outputs which, especially for NLG models, may be computationally prohibitively expensive at test time. Comparing results on a perturbed subset to the full test set would only yield meaningful

| Type | Name | Abbr. | Languages | Communicative Goal | Reference |
|------|------|-------|-----------|--------------------|-----------|
| Data-to-text | CommonGen | CG | en | Expand concept words into natural language | [32] |
| | Czech Restaurant | CR | cs | Verbalize an agents response from dialog act | [33] |
| | E2E | E2 | en | Describe a restaurant from key-value attributes | [34] |
| | ToTTo | TO | en | Describe highlighted cells in a table | [20] |
| | WebNLG | $WN_{en/ru}$ | en,ru | Verbalize subject-predicate-object triples | [35] |
| Text-to-text | MLSum | $ML_{de/es}$ | de, es | Summarize news articles | [36] |
| | Wiki-Auto +TURK/ASSET | $WK_{T/A}$ | en | Simplify a given sentence | [37, 38] |
| | XSum | XS | en | Summarize a news article in one sentence | [39] |
| Dialog | Schema-guided dialog | SG | en | Generate a response given context+dialog act | [40] |

Table 1: Datasets from the GEM benchmark for which we built evaluation suites.

results if the subset was drawn i.i.d. from the test set and given a large enough sample size (since it would be an unbiased sample). We thus support transformations of subsets of the original test set and compare the performance of transformations in relation to the original (or parent) examples. This enables a causal formulation where a perturbation is an intervention and we measure the causal effect of the perturbation on the underlying data. Moreover, while not used in the transformations described in this paper, it also allows us to chain subpopulation identification with transformations, enabling more expressive analyses. The **Data Shift** category spans all test sets that are not directly derived from the original test set. In real scenarios at test time, models will encounter different settings as new data with different characteristics [13]. To evaluate NLG models generalization capabilities in evolving real scenarios, we include test sets bearing a shift in their data distributions. They can be used to measure robustness to time-shift, domain-shift, overfitting, a different language, etc.[2]

## 3.2   NL-Augmenter 🐉→🐍

It is impossible to capture the entirety of natural language phenomena in a few challenge sets and many interesting attributes may be discovered later. It is thus crucial to enable the continuous expansion of existing evaluation suites. In addition, many challenge set construction techniques are reusable. While NLP tasks often radically differ in their linguistic properties of interest — changing the word "happy" to "very happy" in an input is more relevant for sentiment analysis than for summarization — we postulate that many transformations and subpopulations are relevant to many datasets. Thus, the same framework can be used to develop evaluation suites for new datasets.

In this context, we aim to frame the evaluation suite construction problem through open collaboration and we released our framework as a participant-driven repository, called the NL-Augmenter. [3] We invited submissions of *transformation* generators and *filter conditions* for identifying subpopulations which will help expand the challenge sets presented here, but also enable the development of additional evaluation suites. We received 162 submissions from 116 contributors which were reviewed for adequacy and correctness. All participants are asked (i) to address the potential concerns of the reviewers, and (ii) to perform a robustness check of their transformation(s) to assess their relative impact across four models on four datasets.[4] Our framework generates the Transformation sets in a consistent format compatible with existing data loading frameworks like HuggingFace Datasets [31] and we built support for subpopulations directly into our evaluation framework.[5]

## 4   Applying the Framework to Generate Evaluation Suites for 💎 GEM

The Generation, Evaluation, and Metrics benchmark (GEM) [7] aims to identify shortcomings of generation models. The 2021 version features 11 different challenges across 18 languages in data-to-text, text-to-text, and dialog settings. To identify these shortcomings, the benchmark requires evaluation suites which we develop with the framework described in Section 3. Table 1 shows the datasets for which evaluation suites were created. Our selection covers Summarization, Simplification, Data-to-Text, and Dialog. English is the most represented language, but other languages such as

---

[2]This term loosely follows *Dataset Shift* by [13] which includes any non i.i.d. test data, but we distinguish between data derived from existing test cases (transformations) and newly collected data in this category.

[3]`https://gem-benchmark.com/nl_augmenter`

[4]See `https://github.com/GEM-benchmark/NL-Augmenter/tree/main/evaluation` for details.

[5]`https://github.com/GEM-benchmark/GEM-metrics/` (MIT license)

| | | Data-to-text | | | | | | Text-to-text | | | | | Dialog |
|---|---|---|---|---|---|---|---|---|---|---|---|---|---|
| | | CG | CR | E2 | TO | WN$_{en}$ | WN$_{ru}$ | ML$_{de}$ | ML$_{es}$ | WK$_A$ | WK$_T$ | XS | SG |
| SUBPOP | act | | 1 | | | | | | | | | | 1 |
| | frequency | | | | (1) | 3 | 3 | | | | | 1 | |
| | NE feats | | | | 3 | | | | | | | | |
| | shape | | | | | 4 | 4 | | | 1 | 1 | | |
| | size | 1 | 1 | 1 | 2 | 1 | 1 | | | | | | 1 |
| TRANSF | back trans. | | | | | | | | | 1 | 1 | 1 | 1 |
| | typos | | | | | | | | | 2 | 2 | 2 | 2 |
| | no punct. | | | | | | | | | 1 | 1 | 1 | 1 |
| | numbers | | | | | 1 | | | | | | | |
| | order | 1 | 1 | 1 | 1 | 1 | 1 | | | | | | 1 |
| SHIFT | train | 1 | 1 | 1 | 1 | 1 | 1 | 1 | 1 | 1* | 1* | 1 | 1 |
| | validation | 1 | 1 | 1 | 1 | 1 | 1 | 1 | 1 | 1* | 1* | 1 | 1 |
| | time | | | | | | | 1 | 1 | | | 1 | |

Table 2: An overview of the number and types of GEM challenge sets (see full names in Table 1). 1* on the same row are part of the same dataset; (1) means the dataset was already available.

German and Spanish (MLSum), Czech (Czech Restaurants) and Russian (WebNLG) are also present.[6] Table 2 shows the assignment of challenge set types to evaluation suites. In total, we created 80 challenge sets which we further investigate in Section 5.[7]

## 4.1 Subpopulations

The identification of relevant subpopulations relies on the task type, domain-knowledge and familiarity with the data itself and thus varies between datasets. We identify 5 common types of subpopulations, and develop 31 subpopulation sets across them: (i) dialog act type, (ii) frequency in training data, (iii) Named Entity properties, (iv) input shape, and (v) input size.

We split the Czech Restaurant (CR) and Schema-guided Dialog (SG) test data according to the **type of dialog act** present in the input; there are 10 possible acts in SG, and 6 in CR. **Frequency-based** subsets are based on how often certain types of inputs were observed in the training data. For XSum, we look at test subpopulations with different degrees of abstractiveness. To define these subgroups, we use **lexical novelty** of the reference summaries which has been used as a proxy for abstractiveness in previous works [42, 43, 44]. It is computed as the fraction of summary $n$-grams not seen in the input document; in particular here, we compute it as the geometric mean over $n$ =1,2,3. We derive eleven test subgroups with different lexical novelty. Inputs in WebNLG are sets of triples, consisting of a property, subject, and object. For example, the expression Country(paella, Spain) has the predicate *Country*, with subject *paella* and object *Spain*, and communicates that paella comes from Spain. We considered: (a) whether **arguments** were seen or not in the training data (both seen, both unseen, subject unseen, object unseen), (b) whether **single properties** were seen in the training data (seen, unseen), and (c) whether **combinations of properties** in an input triple set were seen in the training data (seen, unseen).We also used the provided frequency-based splits of ToTTo which identify subsets with (un)seen combinations of table column names. We used WikiData to identify three aspects of the **Named Entities** of type *persons* mentioned in ToTTo, which provides information about the Wikipedia page a table originated on: **gender** (*male* and *female*), **nationality grouped by continent** (*Africa*, *Asia*, *Europe*, *North America*, *Oceania*, and *South America*), and **ethnicity** (*African American* and *all USA*); see Appendix D for motivation.

The challenge sets for **input shape** capture the complexity of inputs by characterizing them in task-specific ways. For Turk/ASSET, splits are created in terms of the **syntactic complexity** of the sentences to be simplified. To characterize sentence complexity, we follow Perez-Beltrachini and Gardent [45] and use the 8-level developmental level scale proposed by Covington et al. [46], using the implementation provided by Lu [47] (see scale in Appendix E). For WebNLG (WN), we created subsets based on the **distribution of properties, subject and objects**, with 4 sub-features: (a) maximum number of co-referring subjects (from 0 to 7), (b) presence of co-referring objects, (c) uniqueness of properties, and (d) presence of an entity appearing both as subject and object (*some* and *none* for b-d). Finally, we split the data-to-text and dialog sets based on the **input size**, that is,

---

[6]We exclude the multilingual summarization task WikiLingua [41], for which we had to resplit the data, from the description and evaluations in this paper; we will explore it more in-depth in future work.

[7]Datasets at `https://huggingface.co/datasets/viewer/?dataset=gem` (CC-BY-SA-4.0 license).

the number of properties/dialog acts/highlighted cells in the input: 2 splits for CG, 5 splits for CR, 7 splits for WN and SG, 9 splits for E2E, and 44 splits for Totto. For ToTTo, we additionally split by the total table size (693 different sizes).

## 4.2 Transformations

We apply five types of transformations: (i) back-translation, (ii) typos, (iii) punctuation, (iv) replacement of numerical values, and (v) scrambling of the inputs. As motivated in Section 3, we keep the size of each challenge set to about 500 examples to minimize computational overhead.

We applied perturbations (i), (ii) and (iii) to all English text-to-text and dialog test sets. For **back translation** [48], English input texts were translated to German and back to English[8] using the implementation by Xie et al. [49]. We rejected outputs where the difference in character length between original and transformed example exceeded 35% of the original. For XSum 99.8% of the backtranslations were accepted, for WikiAuto 94.42% (ASSET) and 87.18% (TURK), and for Schema-Guided Dialog 78%. The **typographical errors** (*cheap ->chesp*) were introduced using butter-fingers[9] with two thresholds 0.02 and 0.05, which respectively correspond to lower and higher error frequencies. Finally, **final punctuation signs** were removed if found.

Classifiers are surprisingly robust to shuffling input words [50]. To investigate the same for data-to-text tasks (and Schema-guided Dialog), we randomly change the **order of the components** (triples, concepts, dialog acts, table rows, etc.). Finally, for (iv), we added **numerical variation** in $WN_{en}$, respecting the format of the current cardinal value (e.g., alpha, integer, or floating-point) and replacing it with a new random value. The new number is lower-bounded between zero and upper bounded to be within to the highest power of 10 unit for the given value (e.g., replacing 54 would result in a random value between 0-100). Floating values maintain the degree of precision.

## 4.3 Data Shift

Three types of datasets were created in this category: (i) training data sample, (ii) validation data sample, and (iii) time-Data Shift (25 different types of challenge sets in total). For the **training and validation samples**, 500 data instances were randomly selected in all the training and validation sets. When provided, we tried to match the topic-distribution as the test set. Since the data was seen by a models at training time, we expect an overfitting model to perform better on these challenge sets. This easy-to-create challenge set will be able to provide an indication whether such overfitting is happening. For (iii), we collected **time-Data Shift** to measure how a model responds to context not seen in the training data (and in our case pretraining). For news summarization (MLSum and XSum), we compiled time-shifted test data in the form of articles with Covid19-related keywords.[10] Both datasets were collected before onset of the pandemic and the training data thus does not mention it.

# 5 Experiments

As baselines, we fine-tune publicly available mT5 [51] checkpoints for all datasets presented in Section 4. The mT5 series of models are a multilingual version of T5 [52] - covering 100 languages - which have achieved state-of-the-art results on several multilingual benchmarks. We fine-tune each model for 10,000 steps with a batch size of 262,144 tokens. Following Kale and Rastogi [53], we employ a constant learning rate of 0.001. The best checkpoint is selected based on the BLEU score for the validation set. To study the impact of scale, we experiment with four mT5 variants - Small (300M parameters), Base (600M), Large (1.3B), XL (3.7B). Training is done on 64 TPU v3 [54] chips on an internal cluster.

Our analysis investigates the performance on challenge sets across three types of metrics. (1) **Lexical Overlap**: we use BLEU [55] and ROUGE [56], which score highly when many words overlap between the reference and the generated text; (2) **Semantic Similarity**: we use BLEURT [57] and BERTSCORE [58], which produce a score based on the similarity of embeddings of the reference and the generation. BLEURT is used for English text due to its higher correlation with human

---

[8]See Appendix I for some actual back-translated examples

[9]`https://github.com/alexyorke/butter-fingers`

[10]We used the scripts provided for the re-creation of MLSum and XSum datasets.

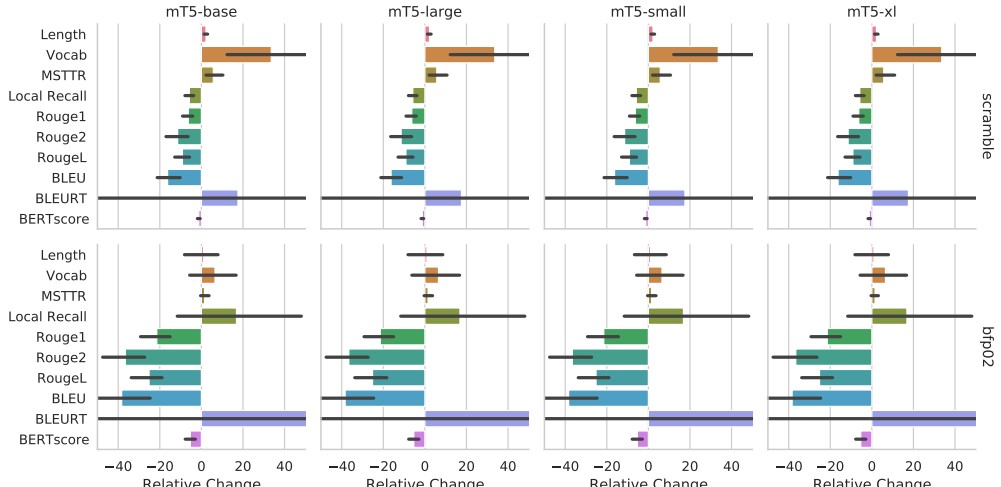

Figure 2: We show the relative metrics change across models as a result of applying the scramble and spelling error transformations. Note that some results are beyond the x-axis cut-off at ±50%.

judgements, and BERTSCORE for all non-English text; (3) **Diversity**: we additionally measure the vocabulary size of all outputs, the average output length (in words), the MSTTR [59], which measures the word-level diversity in the text, and the local recall [60], which identifies which fraction of the different words that appear in exactly one reference are generated by the model.

The results on mT5 variants provide an overview of our higher-coverage evaluation approach compared to only reporting on a single test set. Given the limited space, we omit alternative model architectures, but we are currently using the evaluation suites to study other models submitted to the GEM shared task; the system outputs and scores for the mT5 variants, as well as standard T5, byT5 and other submitted models are already available.[11]

## 6  Analyses and Results

The result of our evaluation is a vast dataset with 56 different metrics × 940 challenge sets and subpopulations, i.e. 52,640 scores per model. We can thus present only a small subset of the possible analyses and anticipate that our framework and the released data encourage others to extend them. While we acknowledge the increased complexity of model evaluation and lack of best practices, we believe that the improved expressiveness of model analyses is worth the additional time investment.

**The effect of input complexity can vary by dataset.** The input length is a strong indication of the difficulty of an example in our datasets, and we thus expect scores to drop with increased input length. Table 3 shows the BLEURT, BLEU, and the prediction lengths on different subsets of the English part of the WebNLG dataset. In line with Shimorina et al.'s earlier results [61], as the number of input properties increases, BLEURT and BLEU decrease while the prediction length increases. Intuitively, more complex inputs make the task more difficult,

| Subset | WebNLG English | | | Czech Restaurants | | |
|---|---|---|---|---|---|---|
| | BLEURT | BLEU | Length | BS | BLEU | Length |
| Val | 0.46 | 66.24 | 21.56 | 0.90 | 17.14 | 10.52 |
| Test | 0.03 | 42.17 | 25.20 | 0.90 | 19.25 | 10.39 |
| 1 prop | 0.20 | 47.34 | 10.47 | 0.85 | 5.34 | 8.14 |
| 2 props | 0.08 | 42.72 | 18.11 | 0.91 | 21.26 | 9.76 |
| 3 props | 0.03 | 40.05 | 25.00 | 0.90 | 20.76 | 11.24 |
| 4 props | -0.04 | 40.18 | 31.86 | 0.92 | 18.49 | 13.71 |
| 5 props | -0.10 | 41.18 | 36.22 | 0.92 | 23.65 | 14.67 |
| 6 props | -0.10 | 41.75 | 42.23 | – | – | – |
| 7 props | -0.14 | 43.24 | 46.20 | – | – | – |

Table 3: BLEURT/BERTSCORE (BS) and BLEU scores, and prediction lengths for the mT5 base model.

and as a result the systems perform worse for inputs with more properties. Additionally, for all models, there is a significant performance drop from the validation to the test set. Considering this initial result, it is surprising to see that the models trained on the Czech restaurant dataset do not demonstrate this behavior. The table also shows that, counter to our expectations, there is no consistent effect on the performance across input sizes. Moreover, the performance gap between the

---

[11]https://gem-benchmark.com/resources

validation and test set is much smaller for this dataset, with no clear differences between the two. The same lack of effect appears for ToTTo where there is no effect when considering either the size of the input table or the number of highlighted cells within. Our extended complexity results in Appendix G demonstrate similar variance in the simplification and dialog sets and for other mT5 models.

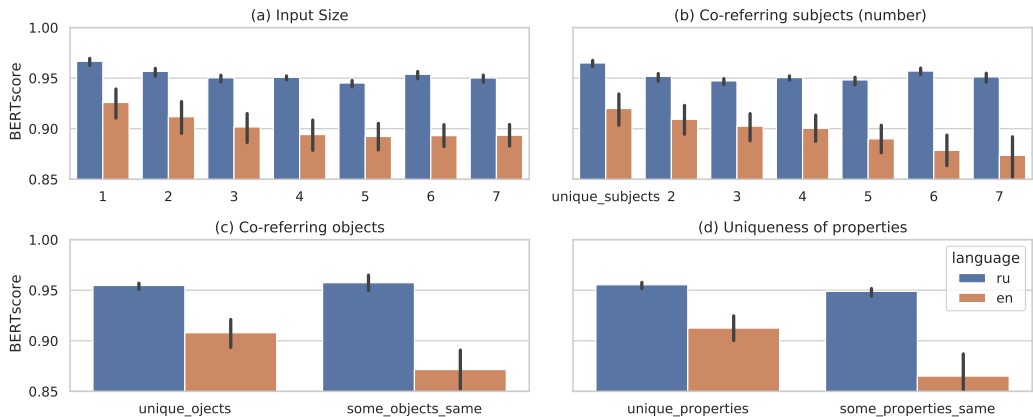

Figure 3: WebNLG results for English and Russian for some subpopulations. The scores of the four models are averaged; error bars indicate variance between model sizes.

**Properties besides complexity influence the model performance.** As established above, our complexity indicator "length" only sometimes leads to insight into a model's performance. However, we can use other subpopulation sets to identify other proxy-attributes. For example, Figure 3(b-d) demonstrates that in WebNLG, whether inputs share subjects, objects or properties has no effect on the Russian models, but a strong one on English. While the difference between languages can be surprising, it is reasonable to assume that it is more challenging to produce a text with pronominalizations, relatives clauses, etc., as needed when entities are shared by several properties. However, there are also unexpected proxies for performance, as shown for ToTTo in Figure 4a. Here, models consistently perform better on Tables that describe male people compared to female, and worse on the African American subset compared to overall US citizens. Moreover, there is a strong variation in performance between people from different continents.

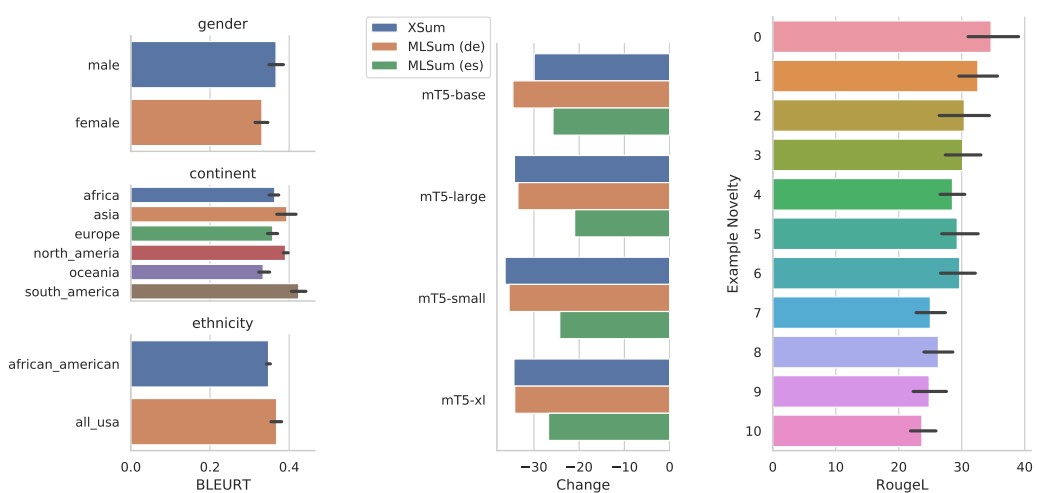

(a) BLEURT results on the fairness-related ToTTo subpopulations.

(b) Relative BERTSCORE change between original and COVID sets.

(c) ROUGE-L score based on fraction of novel words in XSum

Figure 4: The three figures demonstrate the expressiveness of results from challenge sets. In (a), we observe performance differences between subpopulations that talk about people, and in (b) and (c) we demonstrate that models perform poorly when they encounter new concepts and words.

**Test sets vastly overestimate the performance on unseen topics.** So far, the results focus on properties of examples that can be considered within-distribution. They thus do not tell us what happens when a model encounters realistic, yet out of (training) distribution examples. When investigating the performance on the three summarization test sets focusing on COVID-19, shown in Figure 4b, we can see a consistent and sharp decrease in performance.[12] In all three cases, we observe drops of almost 40% compared to the overall test set, indicating that the reported performance number overstates the actual model ability. This can have multiple reasons, ranging from train-test overlap to whether the model was pretrained on topics (or even the data) in the test set. A similar effect can be seen for WebNLG — Both the English and Russian results consistently decrease when aspects of the input were unobserved in the training set.[13] These findings are in line with those by Shimorina et al. [61], and by Søgaard et al. [18] who find that newly collected data commonly leads to worse performance than even adversarial examples. Ngo et al. [62] present similar results for Language Models.

**Transformations uncover overfitting and brittleness.** In Figure 2, we show the effect of different transformations across datasets.[14] As mentioned in Section 4, all our transformations are such that they do not change the expected output and a well-performing model should achieve the same scores before and after each transformation. However, as can be seen, this is not the case and the performance significantly drops in all cases. One curious case is the scrambled input order. Here, the output vocabulary size increases significantly while the input-output overlap (local recall) decreases, indicating that the model relies more on its language model decoder, potentially ignoring the input. Another observation not focused on the model performance itself regards the calibration of learned metrics. While BERTSCORE tends to output scores in the [0.9,1.0] range, BLEURT has a very high variance across datasets. As a consequence, it is unclear what a change of $x$ means for a model. In contrast, lexical metrics like BLEU and ROUGE have much more calibrated changes that are much more alike. Curiously, BLEURT tends to disagree with the other metrics, which could be due to it being the only metric that is trained to detect perturbations similar to the ones we are applying.

# 7 Broader impact

**Overall Limitations** Our goal is to provide a platform with a broad range of different datasets to be able to compare the properties of different system architectures. The presented evaluation suites are neither sufficient nor exhaustive and thus are not a replacement for quality control and human-subject studies.

**Dataset Limitations** Most of the challenge sets are generated based on the original test sets. This means we are limited in the kinds of subpopulations that we can look at, and we are limited in the extent to which the different subsets can be balanced. It is unavoidable that some splits will be too small to be included in the comparison. However, where comparisons are possible, our analyses show interesting differences between models of different sizes, and models trained on different datasets. Still, we do need to be cautious in our interpretation of these results, since our approach leaves room for different confounding factors in the distribution of the inputs. The process of creating subpopulations to test on has also revealed different shortcomings in existing NLG datasets. For example, by exploring different potential subpopulations in the Czech Restaurant dataset, we found that there are no unseen restaurant names or addresses in the test set, which limits the generalizability of the task. These observations provide a motivation for dataset developers to think about challenge sets as part of the dataset design.

**Analysis Limitations** One of the unexplored aspects of this work is the fact that it does not clearly answer whether the results shown in Section 6 are due to inherent limitations in either of the evaluation metrics or the models used. A possible answer is that limitations are within the models. This is due to the fact that most evaluation metrics have been independently explored and validated and are grounded in literature by past researchers more extensively than the models. Therefore, the results presented in this paper show a broader range of performances across multiple dimensions that were previously unknown.

---

[12]While the figure focuses on BERTSCORE, the same effect can be seen on other metrics.
[13]Results shown in Appendix G.
[14]Extended version shown in Appendix H.

Possible future work could help answer this question for any new task and metric combination. This could be done by checking the model's vocabulary for overlap with the test/train corpus using frequency-based metrics for the inclusion of keywords. However, this is a broader problem that goes beyond our evaluation suite.

**Accessibility** Another consideration for the creation of evaluation suites is accessibility, specifically the extent of programming experience required to construct the datasets. Most related approaches construct examples programmatically, which can lead to problems when users instead of developers are asked to assist with evaluation. Projects like SyntaxGym [63] or Dynabench [29] enable the development of targeted test sets through interactive user interfaces. Similarly, BIG bench[15] uses small and often manually curated test sets to assess the capability of large language models to perform on few-shot tasks. While not all of the attribute/dataset combinations we consider are measurable without programming interventions, we allow the manual dataset construction similar to BIG bench.

**Representation** Expanding on the previous point, we further note that the construction of evaluation should in the optimal case include native speakers of a language and be done in consultation with the communities affected by the collected data. The reasons to construct a challenge set, in particular one that connects model performance to protected attributes, should be thoroughly documented.

**Data cards** All datasets used in this work are documented at `https://gem-benchmark.com/data_cards` [64]. We updated the template to add a new header, titled *Changes to the Original Dataset for GEM*. Here we included a list of all subpopulations and perturbations, and known limitations.

**Citation practices** When larger benchmarks combine several smaller datasets, and provide a single overall performance metric, only the benchmarks tend to be cited. Not citing the authors of these datasets means overlooking their creators, which in turn disincentivizes dataset creation. If you use our evaluation suites, we recommend that you also cite the original datasets, and if you use the model outputs, you should also cite the creators of those models.

# 8   Conclusion

In this work, we introduced a framework for the creation of evaluation suites for various natural language generation tasks. We release our code as part of the NL-Augmenter framework which will assist in the creation of similar sets for many different tasks. We use NL-Augmenter to create and release 80 different challenge sets across 12 evaluation suites. Through our analysis of system outputs of four different models, we showcase the kinds of analyses that the evaluation suites enable which can be much more extensive than prior work and of which we can only demonstrate a small selection.

Even in the small selection of results we present, we observe significant variance between results, which means that these performance differences exist and are not typically found and/or reported. With evaluation suites, they can finally be surfaced. We demonstrate clear aggregate patterns across datasets that can be studied more in-depth which is only possible by applying transformations to multiple tasks.

It is important to note that we do not expect anyone to exhaustively explore all possible permutations of metric $\times$ task $\times$ challenge set combinations. Instead, evaluation suites make it easier to address specific questions on model robustness, but also on the effect of, for example, diversity-increasing techniques on model performance.

## Acknowledgments and Disclosure of Funding

Mille's work on this study was supported by the European Commission under the H2020 program contract numbers 786731, 825079, 870930 and 952133. Laura Perez-Beltrachini gratefully acknowledges the support of the European Research Council (award number 681760).

---

[15]`https://github.com/google/BIG-bench`

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
