# OpenReview forum: "Automatic Construction of Evaluation Suites for Natural Language Generation Datasets"
_NeurIPS.cc/2021/Track/Datasets_and_Benchmarks/Round1 — NeurIPS 2021 Datasets and Benchmarks Track (Round 1)_

### Official Review · Reviewer_Xp76 · 2021-07-02
**a principled approach to extending datasets for evaluation**

**Rating:** 9
**Confidence:** 3

**Strengths:**

The paper presents a very interesting approach for extending evaluation of NLP models beyond simple benchmarking.
As the authors admit, using the Subpopulation and Transformation approaches requires the researcher to be intimately familiar with the dataset (which is not always possible, e.g. when the data is multilingual), in order to understand/define whether a specific subsampling or transforamtion makes sense. And of couse Data Shift in is not trivial at all (availability concern). But overall, the basic ideas of the paper are sound and their demonstration is impressive. Even if one disagrees with some of the poins/suggestions/claims made in the paper, the paper itself is a good strating point for discussion, and that is a very good merit in it itself.

**Weaknesses:**

One problem with the approach presented in the paper is that it leads to a 'monstrosity'.
As the authors say in section 6 "The result of our evaluation is a vast dataset with 56 different metrics, 940 challenge sets and 228 subpopulations, i.e. 52,640 scores per model."
Analyzing such an outpoor of results is not easy, and looking for insights can be prohibitive. And it seems to require a non-trivial team to work on just interpreting such results.
Of course, this paper provides just a demonstration of what can be done, and does not require to always produce so much output.
But I think that overall this approach, with all the sound logic (evaluation-wise) does have a tendecy to go towards 'monstrously' big evaluations which can quickly become 'too big to manage'.


**Additional Feedback:**

N/A

**Clarity:**

The paper is sufficiently clear.
Of course, the paper covers a lot of ground,  specifically it mentions many different evaluation sets and tasks, and without some familiarity with at least some of them, it would be very difficult to understand.
However, I think that with a bit of effort, a competent NLP researcher/practitioner can find their way through it.
Also, the paper presents a lot of results, which is not easy to digest in one or two readings. But it is not supposed to be an easy read - and so, given the complexity of the topic, I think it is sufficiently clear.

**Correctness:**

The question whether the dataset is constructed in a sound way is problematic here. The whole point of the paper is to argue that a challenging evaluation suite should be constructed in a certain way. Now, one may agree or disagree with the arguments presented in the paper about the way to construct such a suite, but at least it is obvious that the  suite as presented follows the blueprint that was argued for.

The authors also present results of running their evaluations, and their interpretations of those results. Again, one might argue for different interpretations in some cases, but overall the presentation seems sound. After all, the authors point was not to argue about specific interpretations of experiment results, but rather demosntrate how their approach (and system) facilitate the generation of interesting challenges that might be used for inferring about NLP models. That is certainly achieved.

**Documentation:**

Yes, the dataset is well documented, with data cards, etc.; In fact it is a collection of previosly published datasets, which are 'combined' for the purpose of providing an evaluation suite.

**Ethics:**

The paper presents a section that discusses some  limitations and implications of the research, which is great. I don't find ethical problems with the paper.

**Relation To Prior Work:**

The paper mentions a lot of prior work, especially from recent years - in fact it is based on much prior work, both in the evaluation domain, and the specific datasets that it uses.

**Summary And Contributions:**

The paper describes the logic of constructing an evaluation suite for NLG tasks that purports to go beyond the limitations of simple benchmarking. Given a dataset (or several) that is used for evaluating NLP models on a task (or a set of tasks), the current paper proposes ways of augmenting the dataset in ways that can provide additional insights on NLP model performance.
Specifically, the paper proposes construction of challeging subsets by using three principled approaches. One of them is Subpopulation, which is essentioally different sampling from the original dataset, when such sampling has particular logic for the evaluation. Another approach is data transformation, which basically amounts to corrupting/changing the dataset inputs in some ways (such as typos) - if such transformations can be illuminating for model evaluations. The third approach is called 'Data-Shift' and it is a bit more complex. As the authors say, it amounts to "test sets that are not directly derived from the original test set". Basically, it means using data that is not part of the original dataset - in order to test robustness of the system to use in different circumstances ("evaluate NLG models generalization capabilities"). This is of course not easy - one needs a different dataset for that - for example news from a different time period, or data in another language - and that needs to have some gold standards against which evaluation is done. Often, such datasets are not easy to provide. But logically, it makes sense, and so if there is a way to test a model in different circumstances, it can be very useful for evaluation.

The paper then proceeds with presenting the framework NL-Augmenter that supports the genration of challenge sets for existing datasets. The paper then reports on actual experimentation with this approach generating challenging sets for the GEM benchmark set. The paper then continues to show how  performance of NLP models on such challenge sets can be used to obtain better understaning of the models, their shortcomings or advantages.

---

### Official Review · Reviewer_Lz86 · 2021-07-02
**A framework that can be used to generate challenge test sets for natural language generation**

**Rating:** 6
**Confidence:** 2
**Correctness:** Yes

**Strengths:**

* A detailed study that covers vaiours approaches of generating challenge sets for NLG
* A flatform that enables evaluation suite construction through open collaboration.

**Weaknesses:**

* This work may be more like a tool suite for diagnosis and testing, rather than an evaluation suite. 80 challenge sets across 12 evaluation suites may be useful to identify the potential limitations of different models, but I am not sure how it can be used to compare different models, especially when contradicting findings may usually be observed.
* Only mT5 variants of different sizes are studied, it may be interesting to compare models with different architectures, pretrainng objectives, pretraining data.

**Additional Feedback:**

* How do you distinguish the ineffectiveness of evaluation metrics and models? For example, when the models are evaluated on new topics, is that possible to quantify the ineffectiveness of metrics (BLEURT, BERTScore). Put it another way, is that possible the poor performance is due to the fact BERTScore may fail when it needs to assign a score to sentences containing new concepts or words.

**Clarity:**

* The paper is well written.

**Documentation:**

* There are detailed instructions on Github

**Relation To Prior Work:**

* The authors clearly describe how this work benefits from prior work. However, it is not very clear whether all these subpopulations/transformations have been explored by previous studies, or some of them are tailor for NLG only?

**Summary And Contributions:**

The proposed framework is an extention of RobustnessGym to the generation task.
Given an exisiting test set, the proposed framework can be used to create different types of challenge sets: subpopulation (a subset of test set that focuses on particular features); transformation (a set of examples whose inputs and outputs are modified); and data shift (focusing on a shift, e.g., time, domain, in the data distribution)

A case study is conducted using the proposed framework to generate evaluation suites for GEM generation benchmark, several mT5 variants are studied to demonstrate the usefulness of proposed framework. In other words, different challenge sets are supposed to reveal the weakness of trained models.

In addition to different ways to create challenge sets described in this paper, another important contribution of this work is that it enables open collaboration. That is, a platform (or framework) is built to accept submissions of transformations and conditions for identifying subpopulations

---

### Official Review · Reviewer_TcUS · 2021-07-04
**Well written paper with incremental contributions over the GEM benchmark.**

**Rating:** 7
**Confidence:** 4

**Strengths:**

The main strengths of the paper are the contextualization of this work with the broader literature (Sec 2) and clear descriptions of where the contributions of this paper are (Sec 3). I also appreciated the experimental analysis using the challenge sets (Sec 5 & 6) and the well kept documentation for the datasets included in the benchmark (via Data cards).


**Weaknesses:**

I see two main limitations of this work:
1. The specific contribution of this work is unclear when compared to the [Gehrmann et al GEM benchmark paper](https://arxiv.org/pdf/2102.01672.pdf): Gehrmann et al, largely seems to contain the descriptions of the challenge sets described here. In light of this the major contribution of this paper is the experimental analysis, which brings me to the next point.
2. While the challenge sets and the benchmark seem to provide abundant opportunity for analysis the analysis presented here seems unextensive. While the benchmark might tease apart differences between models the experiments in this paper seemed to have used a single architecture (multilingual T5; while varying it in model size) and most experiments report scores aggregated over these model variants (With Fig 4b & 2 being an exception). I would have loved to see a more nuanced case study based analysis where the utility of the challenge sets may have been demonstrated more clearly. For example, models intended for different end applications could have been shown to differ on some subset of metrics in the approximately 52K metrics the benchmark+challenge sets compute per model. The smallest model tested also contains 300M parameters, I would have liked to see experiments with models of smaller size as well. (BERT base for example contains ~110M parameters)

**Additional Feedback:**

Par 135-141: I appreciate the mechanism to allow additional contributors. Given that tha Gehrmann et al GEM benchmark paper describes in great detail the rationale for inclusion of datasets, what quality control measures will be put in place for the submission of challenge sets?

Table 2: What are the numbers >1 indicating in the cells?

Line 160: How was lexical novelty estimated?

Line 161-164: Consider elaborating on the properties in the body of the submission for clarity.

Line 184-185: Consider describing the kinds of changes back translation makes to the text. While typographic errors, and punctuation errors are error types backtranslation refers more to a mechanism for change than a description of the outcome.

Line 201-203: Would you consider the inclusion of training and validation "challenge sets" as trivial distribution shifts for evaluating models with? I appreciate the time shift as being well motivated but train and validation sets seem trivial.

**Clarity:**

The paper is largely well written though some details seem to conflict between the appendix and the main body of the paper:

Line 212-213: While these lines say that the models were fine-tuned, Lines 534-535 say the benchmarks are test only and no training was carried out. Please clarify.

Line 504-507: Why does this line say the datasets are proprietary and has no licenses included? The Gehrmann et al GEM benchmark paper seems to suggest that the datasets were selected to be openly available.

**Correctness:**

The papers claims seem correct.


**Documentation:**

Concerns are outlined in the "Clarity" sections.

**Ethics:**

No.

**Relation To Prior Work:**

Concerns are outlined in the "Weakness" sections.


**Summary And Contributions:**

The paper mainly summarizes a set of automatically generated challenge sets of the existing GEM benchmark for evaluating language generation approaches, places the challenge sets in the context of the the broader literature, and contributes some  experiments and presents a preliminary evaluation using the GEM challenge sets.

---

### Decision · Program_Chairs · 2021-07-27

**Decision:**

Accept

**Comment:**

The authors introduce a benchmark for evaluating NLG models in a more comprehensive way via challenge sets. The reviewers noted the novelty approach and the quality of its contextualization in the prior literature. There is a consensus on acceptance.